# Comparing endoscopic mucosal resection with endoscopic submucosal dissection in colorectal adenoma and tumors: Meta-analysis and system review

Nian Wang[1◉], Lei Shu[1◉], Song Liu[1], Lin Yang[1], Tao Bai[2], Zhaohong Shi[1]*, Xinghuang Liu[2]*

1 Division of Gastroenterology, Wuhan No. 1 Hospital, Wuhan, China, 2 Division of Gastroenterology, Union Hospital, Tongji Medical College, Huazhong University of Science and Technology, Wuhan, China

◉ These authors contributed equally to this work.
* xh_liu@hust.edu.cn (XL); whyyy_szh@163.com (ZS)

## Abstract

### Aims

This study aimed to evaluate the safety, efficacy, and long-term outcomes of endoscopic mucosal resection (EMR) and endoscopic submucosal dissection (ESD) for treating colorectal adenomas and tumors.

### Methods

A systematic literature review was conducted using databases including PubMed, Web of Science, and Embase. Parameters such as number of patients or lesions, histological diagnosis, lesion size, surgery time, en-bloc resection, R0 resection, severe postoperative complications, and local recurrence were extracted and pooled for analysis.

### Results

A total of 12 retrospective studies involving 1289 patients and 1850 lesions were included in the analysis. EMR was found to have a shorter operation time by 53.6 minutes (95% CI: 51.3, 55.9, P<0.001) and fewer incidences of severe postoperative complications such as perforation and delayed bleeding (OR = 0.40, 95%CI: 0.23, 0.71, P<0.001). On the other hand, ESD had higher rates of en-bloc resection (OR = 0.15, 95%CI: 0.07, 0.30, P<0.001) and R0 resection (OR = 0.32, 95%CI: 0.16, 0.65, P<0.001). Recurrence after EMR was found to be significantly higher than that after ESD surgery (OR = 5.88, 95%CI: 2.15, 16.07, P = 0.037).

### Conclusions

The study suggests that the choice of surgical method may have a greater impact on recurrence compared to the pathological type, and that ESD may be more suitable for the

**Data Availability Statement:** All relevant data are within the paper and its Supporting information files.

**Funding:** This work was funded by the Natural Science Foundation of Hubei Province (No.2020CFB358) and Wuhan Health Commission Youth Fund (No.WZ22Q29). The funders had no role in study design, data collection and analysis, decision to publish, or preparation of the manuscript.

**Competing interests:** The authors declare no conflict of interest.

treatment of malignant lesions despite its higher rates of severe postoperative complications and longer operation time.

## 1. Introduction

Cancer is a growing concern globally, and its incidence and mortality rates are rapidly increasing. Colorectal cancer (CRC) was estimated to account for around 10% of all new cancer cases in 2018, totaling around 1.8 million cases [1]. Additionally, CRC related deaths were projected to account for about 9% of cancer related deaths, totaling around 860,000 deaths worldwide. The incidence of colon and rectal cancer is predicted to increase significantly by 2030, especially among younger age groups. The incidence rate is expected to increase by 90.0% and 124.2% among patients aged 20 to 34 for colon and rectal cancer respectively, and by 27.7% and 46.0% among patients aged 35 to 49 [2]. These projections highlight the urgent need for effective screening and treatment of CRC.

It is important to detect CRC early to increase patient survival rates, and screening tests such as colonoscopies have proven to be effective in detecting and preventing CRC. Colonscopic removal of adenomatous polyps can significantly decrease the risk of CRC [3]. In addition, many guidelines recommend endoscopic resection as an effective treatment for colo-rectal lesions, including different resection methods [4–6]. These approaches have proven to be beneficial in reducing morbidity and mortality rates associated with CRC.

Endoscopic mucosal resection (EMR) and endoscopic submucosal dissection (ESD) are minimally invasive and effective techniques for the treatment of precancerous lesions and early stages of CRC. These procedures are widely accepted in medical practice and have shown promising results in treating various types of neoplasia. However, the choice of the procedure depends on various factors such as the size, location, and nature of the lesion, as well as the patient's overall health and medical history.

EMR is a highly effective and minimally invasive technique used to remove precancerous lesions that are less than 2 cm in size in an en-bloc fashion [7]. Utilizing underwater EMR has shown to produce better outcomes in terms of en-bloc and R0 resections and procedure time, particularly for lesions that are within the range of 30–40 mm in colorectal polyps, when compared with conventional EMR [8]. However, it should be noted that EMR is not without its limitations, as the piecemeal resection method is associated with high recurrence rates.

Compared with EMR, ESD achieved a higher R0 resection rate but more time consuming [9]. Moreover, ESD is more effective for early CRC and precancerous lesions, with higher en-bloc resection, higher R0 resection rate and lower local recurrence [9–11]. However, ESD had a higher risk for complications such as perforation and bleeding, while its hospitalization costs were higher, and endoscopic operation was technically demanding and more time consuming [9, 12].

Considering the strengths and weaknesses of both EMR and ESD, there is currently no definitive consensus on the ideal endoscopic therapy for colorectal adenoma and tumor. While previous Meta-analysis studies have been conducted [13–16], they suffer from certain limitations, including the inclusion of conference abstracts, the absence of Chinese literature, and outdated data due to the publication of new articles. Thus, the present study aims to comprehensively analyze and compare the effectiveness and safety of EMR and ESD as treatments for colorectal adenoma and tumor.

## 2. Materials and methods

### 2.1. PICO statement

The PICO statement was following:

- P-patient, problem or population: Patients undergoing endoscopic surgery (EMR or ESD) for colorectal adenomas or cancers.

- I-intervention or exposure: The endoscopic surgery (EMR & ESD).

- C-comparison, control or comparator: Comparison of the safety and efficiency of the two endoscopic procedures, and comparison of patients' prognosis undergoing different endoscopic procedures.

- O-outcome: The primary outcome was the odds ratio (OR) of EMR as a risk factor and ESD as a contrast for the recurrence, the complication and successful excision of the lesion. The secondary outcome was mean differences between EMR and ESD group in terms of surgery time and lesion size.

### 2.2. Search strategy

Our aim was to identify all studies evaluating the difference between EMR and ESD measured by safety, surgical excision efficiency and long-term outcome of the patients. In accordance with the preferred reporting items for systematic reviews and Meta-analyses (PRISMA) guidelines, we searched published studies from PubMed, Web of Science and Embase from inception to September 12, 2022. We included in our analysis only studies that were published in full text or accepted for publication in indexed journals.

We identified all pertinent literature that were associated with endoscopic mucosal resection (EMR), endoscopic submucosal dissection (ESD), colorectal cancer (CRC) and colorectal adenomas (CRA). We used (("Colorectal Adenomas") OR (("Colorectal tumors") OR ((lst) OR ("laterally spreading tumor")))) AND (((EMR) OR ("Endoscopic mucosal resection")) OR (("Endoscopic mucosal dissection") OR (ESD))) as the retrieval strategy. No additional retrieval restrictions were set to achieve higher recall. We also hand-searched the reference sections of previous Meta-analyses for relevant articles. The search was performed by two authors (N.W and XH.L) until no new records could be found.

### 2.3. Inclusion and exclusion criteria

Studies were identified based on predefined inclusion criteria:

a) the study was designed as a case-control, cohort study, cross-sectional study or clinical trail; b) the study declared clear diagnosis criteria with colorectal adenomas OR colorectal tumors OR laterally spreading tumor; c) the study compared the EMR and ESD and had accurate digitized results.

Studies were excluded based on the criteria:

a) the study type was review, Meta-analyses, comments, letters, case reports, non-human studies or meeting abstract; b) duplicate literature would be excluded; c) small sample research (total sample < 30); d) articles that cannot be found in full text would also be rejected.

We extracted the following data from each eligible article:

a) basic information: article title, name of the first author, publication year, country, study design; b) Study related data (numbers were recorded according to two different endoscopic procedures): patient number, age, the proportion of genders, lesion size, surgery time, number

of en-bloc resection, number of R0 resection, number of severe postoperative complications (perforation, bleeding and so on), and number of tumor recurrence.

The records management was performed by two authors (N.W. and XH.L.). Conflicts regarding the inclusion or exclusion of articles were resolved by consensus with a third investigator (T.B.). The software Endnote X9 was used.

## 2.4. Risk of bias assessment

We predicted that the types of literature studies to be included would include non-randomized controlled trials and retrospective analysis, so we chosen the evaluation tools of non-randomized controlled trials, which is more rigorous. The Risk Of Bias In Non-randomized Studies of Interventions (ROBINS-I) tool, which was recommended by the Cochrane Handbook for Systematic Reviews of Interventions to assess the risk of bias in a non-randomized study, was adopted [17, 18]. Non RCT studies were judged for the following seven domains: confounding bias, selection bias, bias in classification of interventions, bias in deviation from intended interventions, bias due to missing data, bias in measurement of outcome and bias in selection of the reported results. All risk of bias assessments were conducted by two reviewers (N.W. & XH. L.). Disagreements were resolved by recruiting a third author (T.B.) to attain consensus.

Studies were not excluded a prior based on quality assessment.

## 2.5. Data extraction

The following data were collected: basic information (title, author, date of publication), sample size, surgical lesion size, surgery time, number of en-bloc & R0 resection, number of severe postoperative complications and number of recurrence for different surgery (ESD or EMR).

In this work, postoperative perforation and delayed bleeding were defined as the serious postoperative complication and the total number of the two occurrences is added. Clinically, there are some derivations of EMR and ESD such as EPMR, EMR-P, CSI-EMR, hybrid ESD, ESD-S and so on. Considering that they were all performed on the basis of EMR or ESD, they would not be separately classified in our study, but included in the EMR or ESD category.

## 2.6. Data synthesis and meta-analysis

In this Meta-analysis, all statistical analyses were performed with STATA 14.0 (Stata Corporation, Texas, USA). Data from selected studies were extracted into 2×2 tables. Relative risk/odds ratio and 95% confidence intervals (95% CI) were calculated, and effect size (weighted mean difference) Meta-analysis was performed. The mean differences, as well as 95% confidence, intervals between EMR and ESD group in terms of surgery time and lesion size were also pooled. The heterogeneity among all the included studies was assessed using $I^2$.

In consideration of that the clinical size of the lesion may affect the operation time and post-operative complications, and the pathological nature of the lesion may affect the recurrence of the patient after surgery, Meta-regression and baseline correction based on Trowman method were performed [19, 20]. Both fixed effect model and random-effect model were applied for Meta-analysis.

Publication bias was assessed by using R-based Robvis software package introduced by the National Institute for Health Research (NIHR) [21].

The ethics approval was not required for the Meta-analysis since this study used only the data from already published studies and did not have patient personal information.

## 3. Results

### 3.1. Articles selection

We retrieved a total of 2,117 records, with 1,061, 380, and 676 records from Embase, PubMed, and Web of Science, respectively. After removing duplicates, we were left with 1,625 records. We screened the abstracts and found that 1,180 records were unrelated to the topic and 412 records did not meet the inclusion criteria for literature type. After the preliminary examination, only 33 records passed. We also found seventeen meeting abstracts, as well as one Korean article which could not be included. Besides, we excluded two articles which were unrelated to our topic, and one article which only had an abstract. After reading the full text, we finally included 12 articles in our work [23–34]. Our procedure is summarized in a preferred reporting item for systematic review and meta-analysis flow diagram (Fig 1).

### 3.2. *Methodology quality assessment and d*ate *extraction*

The quality assessments of the included trials by using ROBINS-I tool were summarized in Table 1 and Fig 2. Most of literature was not clinical trials, so using this evaluation tool was somewhat harsh for most of the literature included. However, this evaluation tool more clearly showed the possible bias of each study, so it was still used.

   After completing the assessment, the relevant data of all twelve articles were extracted, as shown in the Table 2.

### 3.3. Merging and meta-analysis

All the above literatures with purposeful effect values were included in the Meta-analysis, and the effect values were pooled, respectively. EMR was used as the intervention group and ESD was used as the control group.

   **3.3.1. Lesion size and surgery time.**   Clinically, doctors may prefer to choose EMR surgery for small lesions. The study reached similar conclusions that the lesions targeted by EMR tended to be smaller than those treated by ESD (Fig 3a). The tumor size of EMR was 11.0 [95% CI(10.8, 11.2)] millimeters smaller than that of ESD on average.

   EMR is considered to have a shorter operating time because of its simplicity. It prove that the operation time of EMR was 53.6 [95%CI(51.3, 55.9)] minutes shorter than that of ESD on average (Fig 3b).

   Could the difference in lesion size explain the difference in surgical time? We used Meta-regression analyses (Trowman method) to include tumor size as a co-variate and found that the operation time of EMR was still 37.6 minutes shorter than that of ESD on average (P = 0.001) (S1 Table).

   **3.3.2. Histological diagnosis.**   The pathological characteristics of the lesion often greatly affect the physical properties (stiffness, degree of edge smoothness and so on) of the lesion and the prognosis of the patient after resection (recurrence). Physical properties may affect surgical resection, and malignant lesions may predict a poor prognosis.

   In this work, we plotted the number of adenomas and cancers (including carcinoma in situ) treated by EMR, and the number of adenomas and cancers (including carcinoma in situ) treated by MR In a four-grid table, and calculated the OR value. So a higher OR in a given study meant a higher proportion of adenomas in subjects underwent EMR and a higher proportion of cancer in subjects underwent ESD.

   We demonstrated that researchers would choose ESD more for malignant lesions (Fig 4), this probably because the difference between ESD and EMR surgical methods. The pooled OR for histological diagnosis were 1.68 [95%CI(1.20, 2.17)] for random effect model and 1.66

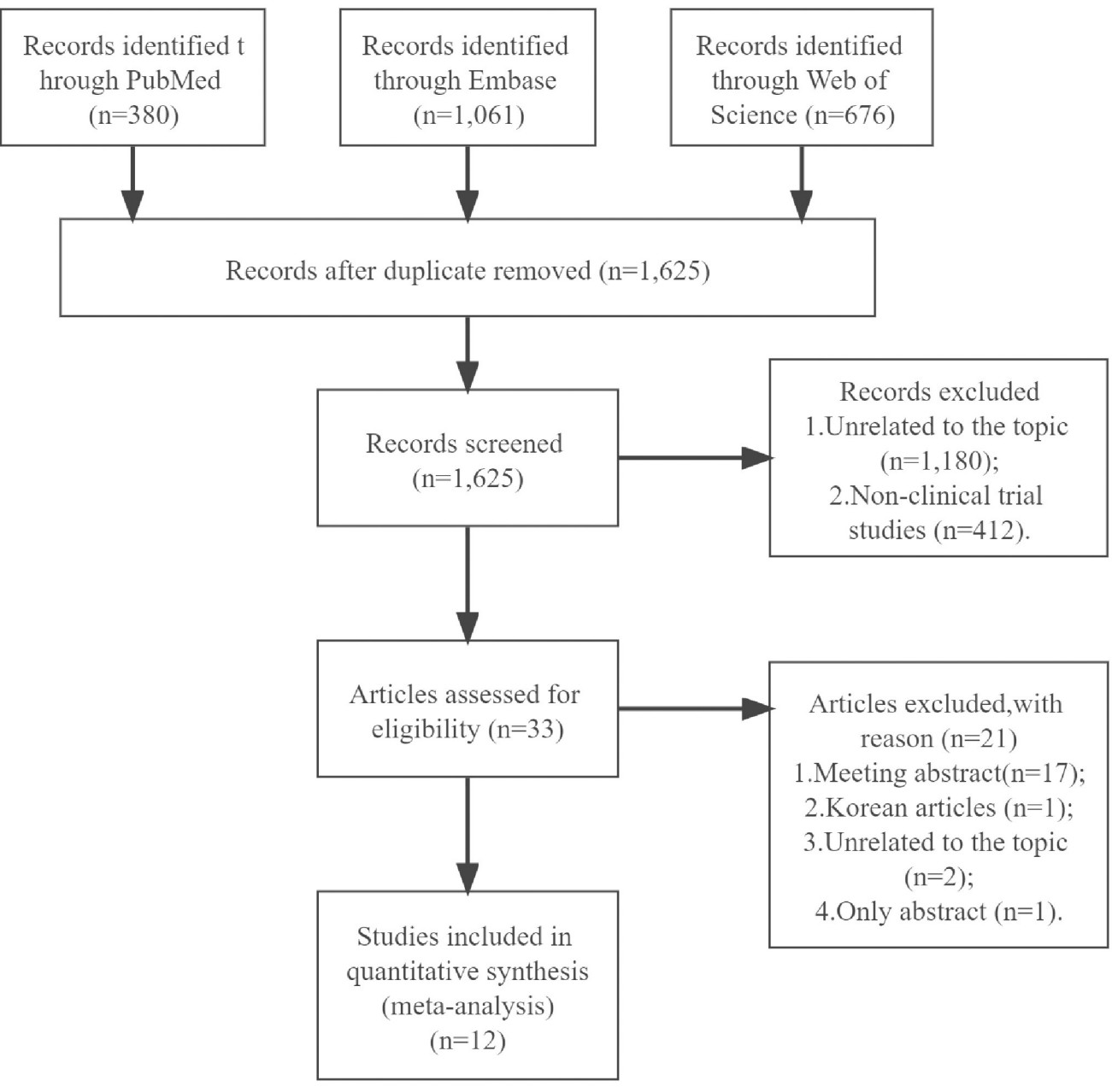

**Fig 1. Flow diagram of inclusion of studies.**

[95%CI(1.22, 2.10)] for fixed effect model. $I^2$ for random effect model was 9.4% and P value was less than 0.001.

**3.3.3. En-bloc resection and R0 resection.** En-bloc resection and R0 resection often predict surgical success and a relatively good prognosis [34]. In our work, we found that EMR had fewer en-bloc resection and R0 resection than ESD. The pooled OR for en-bloc resection were 0.15 [95%CI(0.07, 0.30)] for random effect model and 0.14 [95%CI(0.11, 0.18)] for fixed effect model (Fig 5a). $I^2$ for random effect model was 83.9% and P value was less than 0.001. At the same time, the pooled OR for R0 resection were 0.32 [95%CI(0.16, 0.65)] for random effect

**Table 1. The quality assessment for the included articles.**

| Study | Domain | | | | | | | Overall risk of bias judgements |
|---|---|---|---|---|---|---|---|---|
| | 1 | 2 | 3 | 4 | 5 | 6 | 7 | |
| Yutaka Saito et al. [22] | C | L | L | S | L | L | M | S |
| Yun Jung Kim et al. [23] | C | L | M | S | L | L | M | S |
| Masahiro Tajika et al. [24] | C | M | L | S | L | L | M | S |
| Motomi Terasak et al. [25] | C | L | L | S | L | L | M | S |
| Zou Jiale et al. [26] | C | M | L | S | L | L | M | S |
| Eun-Jung Lee et al. [27] | S | S | L | S | L | L | M | S |
| Yue Li et al. [28] | C | M | L | S | L | L | M | S |
| Nozomu Kobayashi et al. [29] | C | S | L | S | M | L | L | S |
| Dae Young Cheung et al. [30] | M | L | L | L | L | L | L | M |
| Jin-Sung Jung et al. [31] | C | S | L | S | L | L | M | S |
| Takashi Toyonaga et al. [32] | C | S | L | S | L | L | M | S |
| Yongkang Cui et al. [33] | C | M | L | S | L | L | M | S |

The 1–7 domains in the above table represented ①confounding bias, ②selection bias, ③bias in classification of interventions, ④bias in deviation from intended interventions, ⑤bias due to missing data, ⑥bias in measurement of outcome and ⑦bias in selection of the reported results, respectively. Low, moderate, serious, critical risk of bias and no information were simply L, M, S,C and NI respectively.

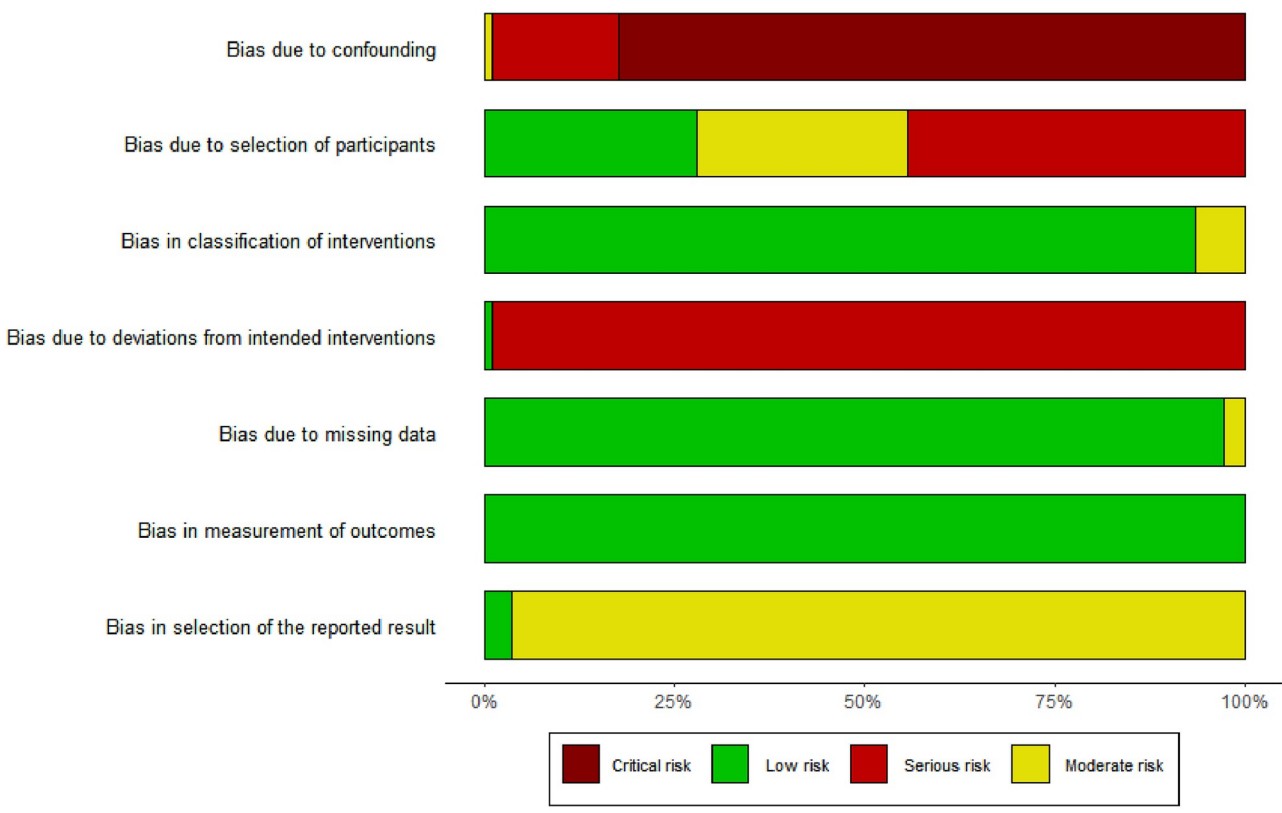

**Fig 2. Summary barplot of standard risk-of-bias.**

**Table 2. Information extracted from the included articles.**

| Study | Number of patients or lesions | Histological diagnosis (benign and malignant) | Tumor dimension (mm) | Surgery time (min) | En bloc resection | R0 resection | Severe postoperative complications | Recurrence |
|---|---|---|---|---|---|---|---|---|
| Yutaka Saito et al. [22] | 228/145 | - | 28±8/37±14 | 29±25/108±7 | 74/122 | - | 10/11 | 33/3 |
| Yun Jung Kim et al. [23] | 91/115 | (55,36)/(50,55) | 20.9±7.9/28.5±10.4 | 30±30.5/64.9±43.3 | 56/93 | 47/75 | 6/31 | 0/1 |
| Masahiro Tajika et al. [24] | 104/85 | (81,23)/(31,54) | 25.5±6.8/31.6±9.0 | 29.4±26.1/87.2±49.7 | 50/71 | 41/71 | 3/7 | 16/1 |
| Motomi Terasak et al. [25] | 178/89 | (128,50)/(51,48) | 32.2±15.6/38.8±17.3 | - | - | - | 18/9 | - |
| Zou Jiale et al. [26] | 52/94 | (51,1)/(89,5) | 1.8/3.0 | 12.8/38.9 | 38/77 | 35/75 | 1/9 | - |
| Eun-Jung Lee et al. [27] | 209/314 | (150,42)/(192,119) | 22.3±4.4/28.9±12.7 | - | 105/291 | 87/275 | 4/27 | 31/2 |
| Yue Li et al. [28] | 160/288 | - | 19.8±1.0/31.1±0.8 | 38.5/98.3 | 155/276 | 28/26 | 2/31 | 14/10 |
| Nozomu Kobayashi et al. [29] | 56/28 | (16,40)/(8,20) | 25.0±9.0/27.1±10.1 | 11/140 | 21/27 | - | 1/5 | 12/0 |
| Dae Young Cheung et al. [30] | 16/17 | - | 6.6±2.0/7.5±1.9 | 9.7±3.6/20.2±12.6 | - | 9/15 | 1/1 | 0/0 |
| Jin-Sung Jung et al. [31] | 127/119 | (62,65)/(49,70) | 23.1±8.8/34.3±11.6 | 20.4±21.1/55.5±41.1 | 82/119 | 107/102 | 19/12 | - |
| Takashi Toyonaga et al. [32] | 24/512 | - | 20.0/28.9 | 19.0/57.2 | 20/503 | - | 0/17 | - |
| Yongkang Cui et al. [33] | 44/44 | (38,6)/(34,10) | - | 23.7±6.19/65.1±13.8 | 26/43 | - | 4/7 | 14/5 |

EMR and ESD data were displayed in the form of EMR/ESD. When there was no relevant data in the article, horizontal lines (-) were used instead. Continuous variables were presented as mean plus or minus one standard deviation.

model and 0.28 [95%CI(0.22, 0.35)] for fixed effect model (Fig 5b). $I^2$ for random effect model was 87.6% and P value was less than 0.001. So ESD can achieve more *en-bloc* resection and R0 resection.

**3.3.4. Severe postoperative complications.** In this study, postoperative perforation and delayed bleeding were counted as serious postoperative complications, and the number of times they occur were added up to get the number of postoperative complications. As Fig 6, the diamond was on the left of the vertical line and did not intersect with the line (OR = 0.40 [95%CI(0.23, 0.71)] for random effect model and OR = 0.42 [95%CI(0.31, 0.57)] for fixed effect model). $I^2$ for random effect model was 58.3% and P value was 0.006. EMR had fewer severe postoperative complications than ESD. The severe postoperative complication rates after EMR and ESD were about 3%-5% and about 7%-10%, respectively (S1a and S1b Fig).

**3.3.5. Recurrence.** Whether there will be a recurrence of adenoma and adenocarcinoma after endoscopic resection is a concern. So, we drew a forest map to explore the question of which endoscopic procedure would reduce recurrence (Fig 7a). One article was excluded because the number of recurrences in both ESD and EMR were zero. We found that the diamond was on the right of the vertical line and did not intersect with the line (OR = 5.88 [95% CI(2.15, 16.07)] for random effect model and OR = 6.39 [95%CI(3.82, 10.68)] for fixed effect model). $I^2$ for random effect model was 57.7% and P value was 0.037. This meant the value of

(a)

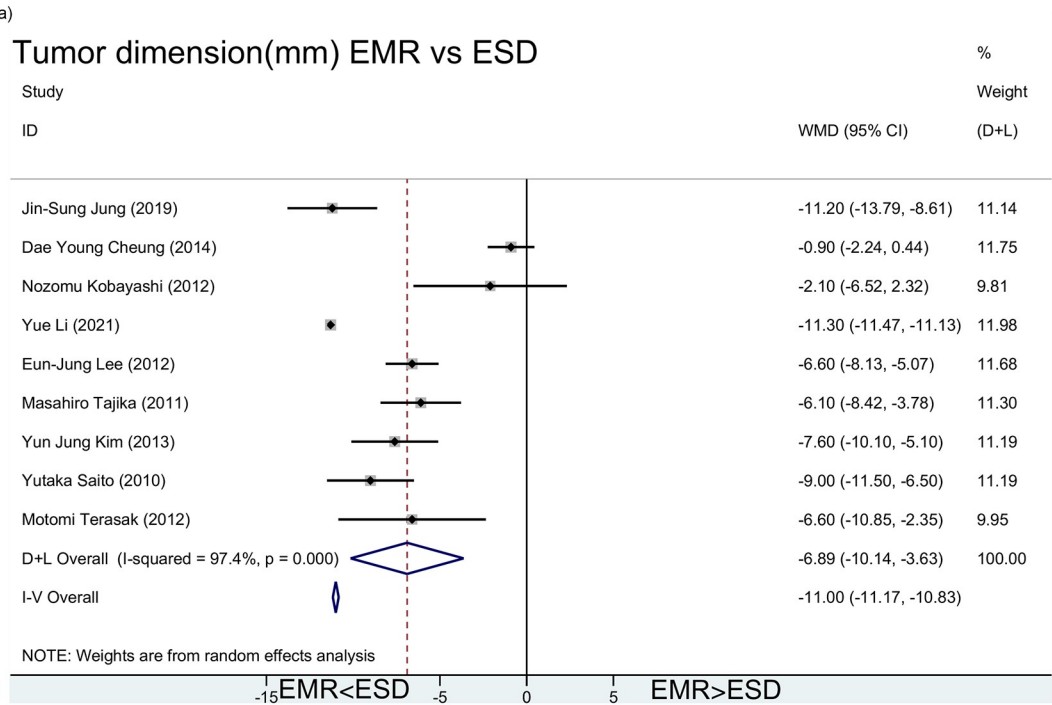

Fig 3. a. Meta-analysis of tumor dimension(mm) difference between EMR and ESD. b. Meta-analysis of surgery time (EMR vs ESD).

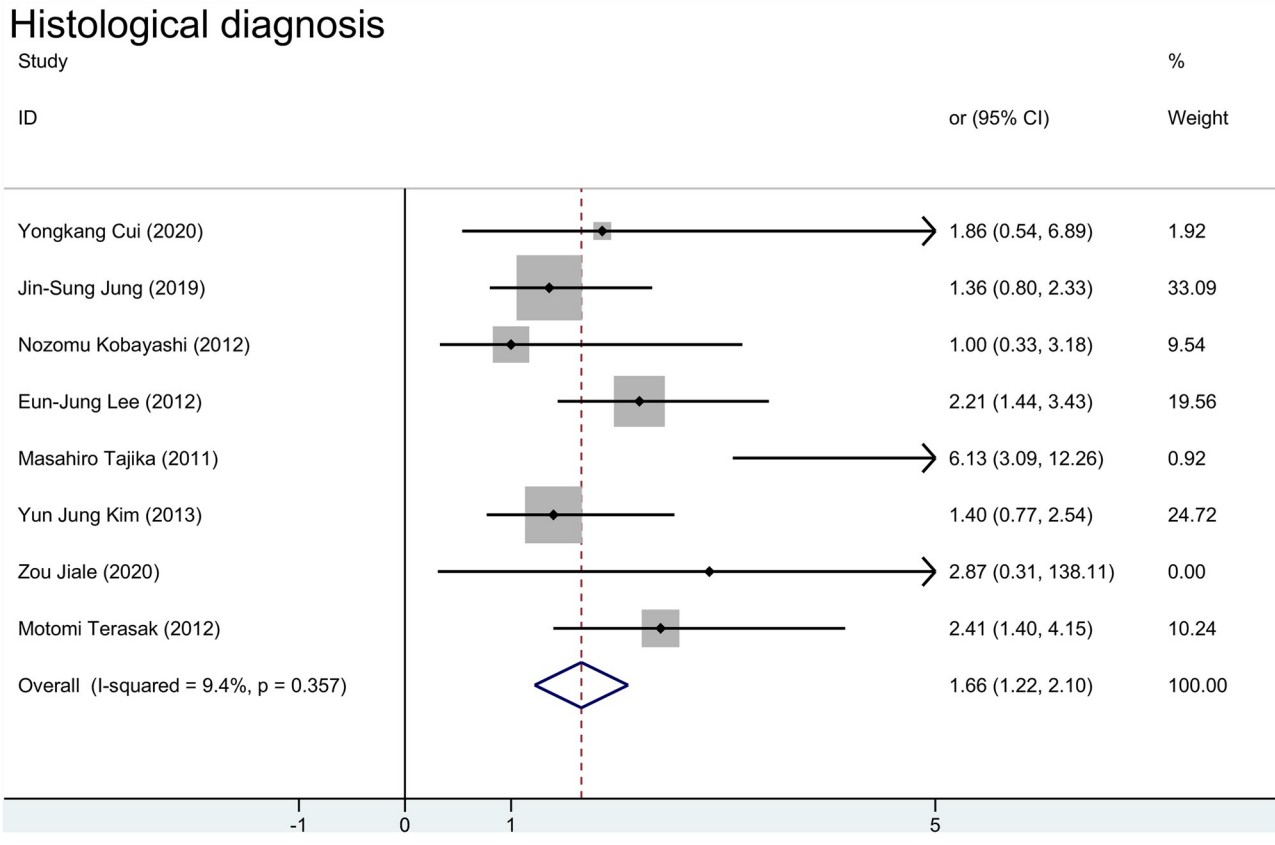

**Fig 4. Meta-analysis of EMR and ESD for histological diagnosis.**

OR was statistically significant. Further, this meant that patients who underwent EMR were more likely to have a recurrence than those who underwent ESD.

The recurrence rate after EMR and ESD were taken as the effective value respectively (Fig 7b and 7c). The average recurrence rates were obtained as follow: about 14–16% for EMR and about 1–2% for ESD.

Using regression analysis, with tumor size as a co-variate, a potential linear relationship between tumor size and recurrence was found, although the results showed no significance (P = 0.291 & P = 0.147 for EMR and ESD, respectively) (S2a and 2b Fig).

Usually, malignant lesions are associated with a poorer prognosis than benign ones. The results showed that the ESD group had a higher percentage of malignant lesions but a lower recurrence rate than EMR. To further confirm the relationship between the lesion type and recurrence, the Meta-regression analysis was adopted (this OR as a co-variate and OR of EMR vs ESD for recurrence was used as the dependent variable). We found this regression meaningless (P = 0.594), and even the number of $tau^2$ increased from 0.8016 to 1.237. This may mean that pathological type has a weaker effect on prognosis compared with surgical method.

## 4. Discussion

This Meta-analysis is the latest comparison of clinical outcomes between ESD and EMR in the treatment of colorectal tumors. A total of 12 articles with 3139 lesions were included in the analysis. The findings indicate that ESD was more effective than EMR in en-bloc resection rate

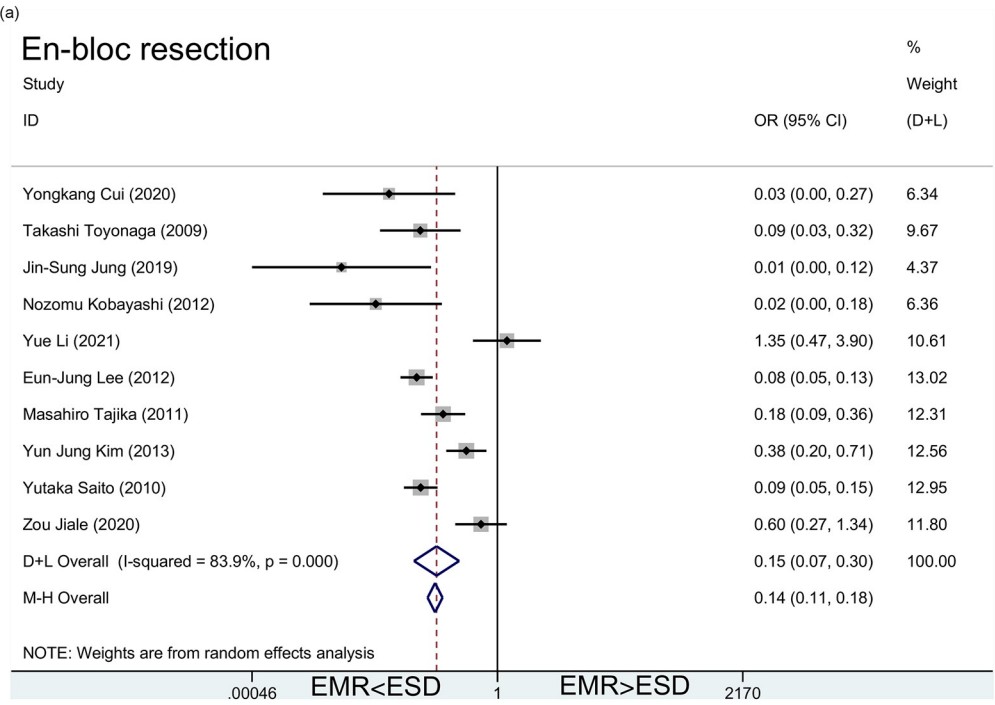

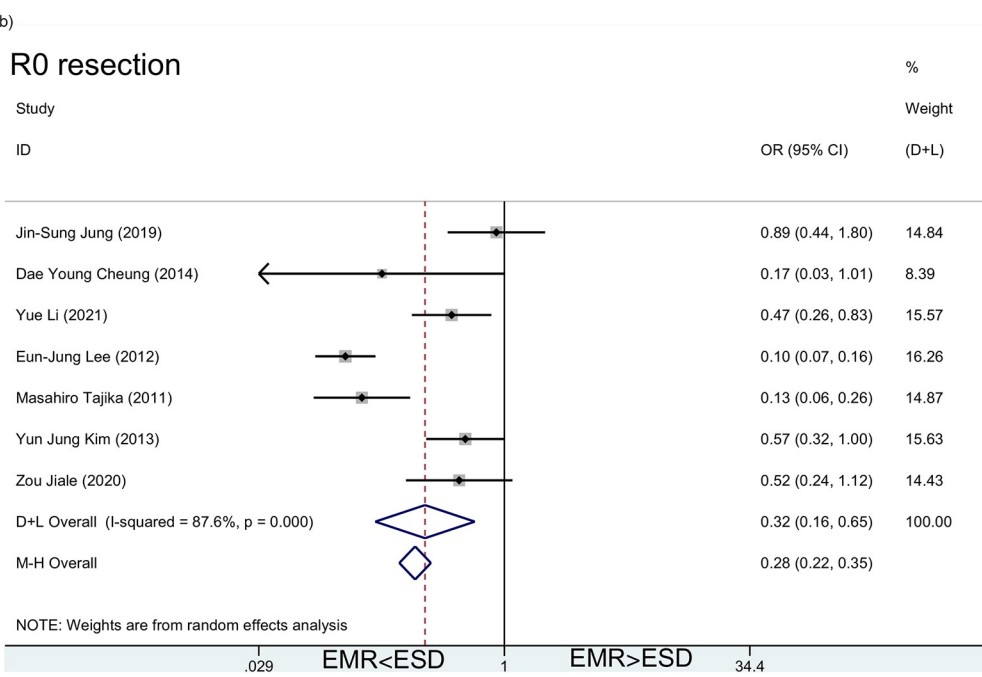

**Fig 5.** a. Meta-analysis of en-bloc resection (EMR vs ESD). b. Meta-analysis of R0 resection (EMR vs ESD).

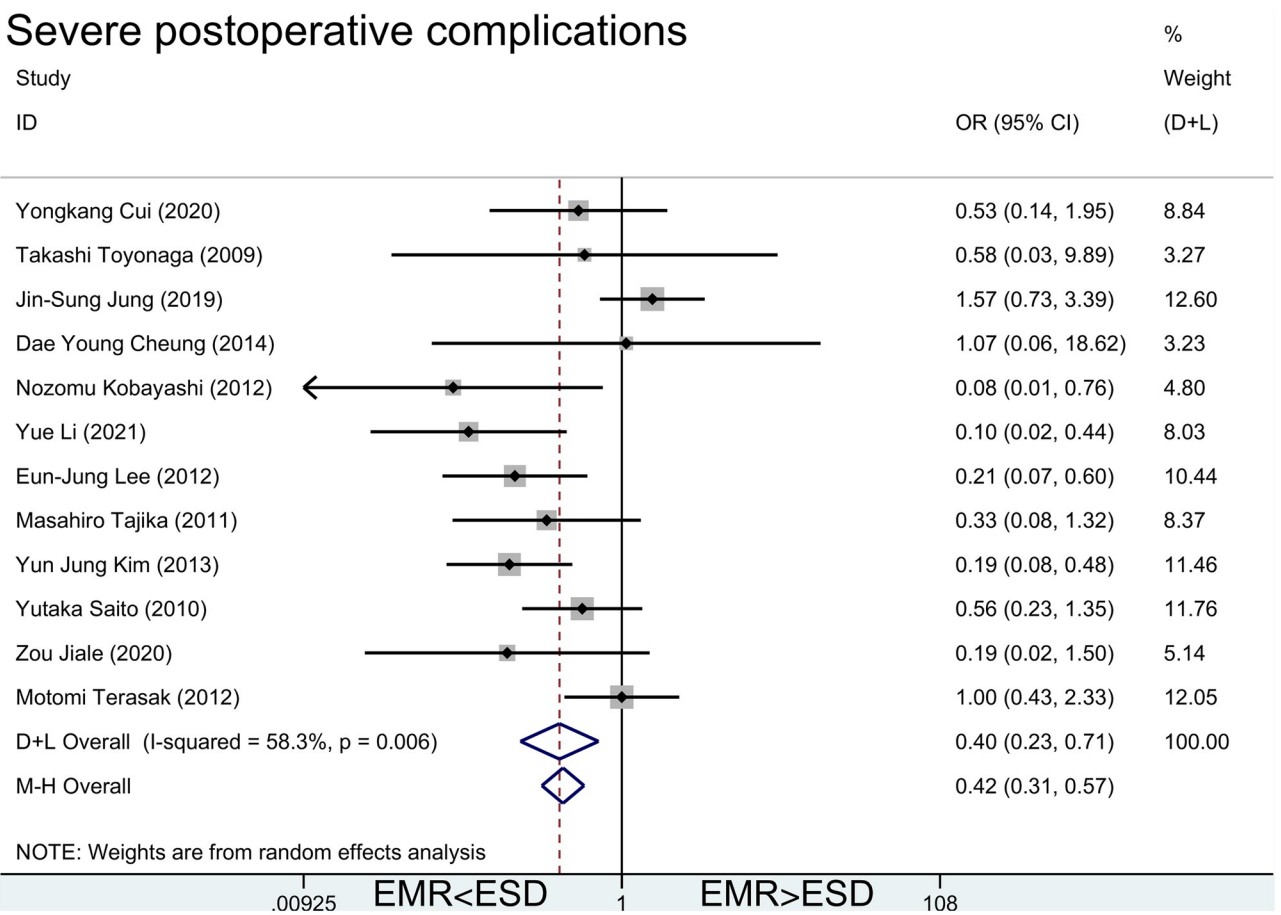

**Fig 6. Meta-analysis of severe postoperative complications after EMR and ESD.**

and R0 resection, with EMR showing lower rates in both. Additionally, the local recurrence rate of EMR was higher than that of ESD. Further analysis revealed that local recurrence was affected by the choice of endoscopic resection method, rather than the postoperative pathological type. EMR had fewer complications, such as delayed bleeding and perforation, compared to ESD. On the other hand, ESD was more time-consuming and was mainly used for larger tumors.

Due to insufficient data, this paper did not conduct separate analyses on ESD and EMR surgery for certain important types of benign and malignant intestinal tumors, such as Colorectal laterally spreading tumors (LSTs) and Neuroendocrine tumors (NETs).

LSTs are flat lesions that grow laterally along the intestinal wall, with a diameter greater than 10 mm [35]. Due to their high risk of malignancy, LSTs are considered pre-cancerous lesions of CRC [36]. LSTs can be subclassified into granular types (LST-G), which have nodular surfaces, and non-granular types (LST-NG), which have smooth surfaces [37]. Early diagnosis and treatment are therefore crucial in preventing the progression of this disease. As LSTs grow laterally along the intestinal wall, they typically have a lower risk of submucosal invasion, making EMR and ESD the preferred initial treatment methods [38].

However, clinical research showed that LST-NG had a higher proportion of submucosal invasive lesions. Therefore, treatment should take into consideration LST subtypes and

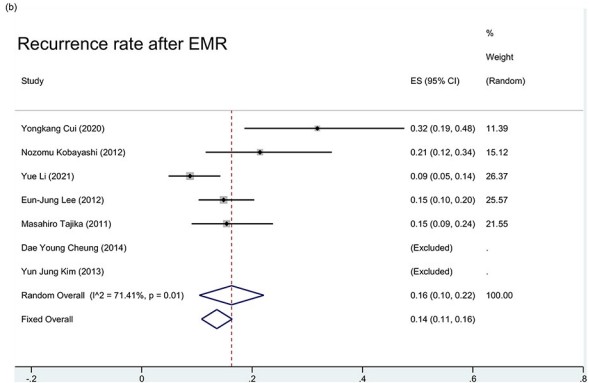

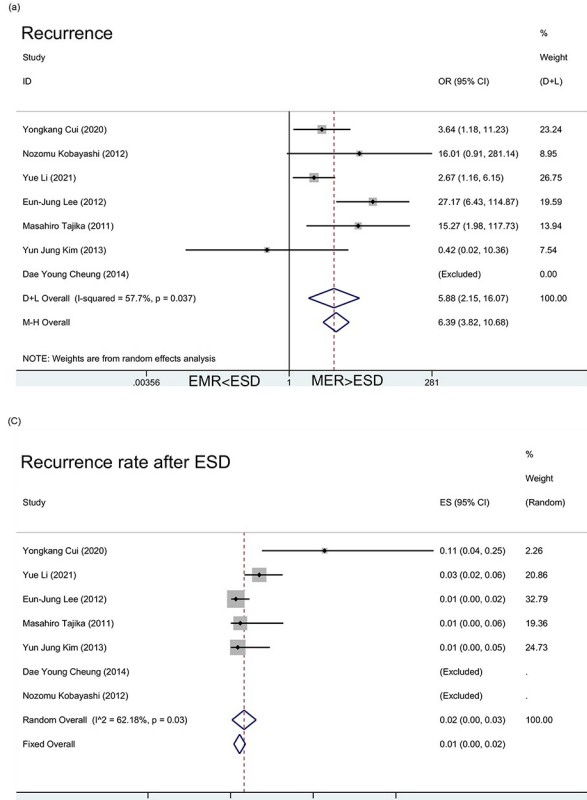

**Fig 7.** a. Meta-analysis of recurrence after EMR and ESD. b. Recurrence rate after EMR. c. Recurrence rate after ESD.

diameter, as well as the quality of life and prognosis of patients [39]. For larger size colorectal LSTs, with nodular mixed subtype, cancerous pit pattern, or early adenocarcinoma, en bloc or curative resection is recommended. ESD is a preferred treatment method [27, 31]. For lesions with a diameter greater than 20mm, EMR cannot achieve en bloc resection, which may result in local residual lesions or recurrence [27]. However, recent studies have shown that EMR is a safe and effective operation for LSTs larger than 4cm, and its effectiveness is not inferior to that of ESD [40]. In earlier research, EMR was considered a first-line treatment for LSTs, with a low risk of submucosal invasion [38]. For LSTs, the en-bloc resection rate of ESD and EMR were 96% and 93.7%, respectively, and the corresponding pathological R0 resection rates were 90.1% and 82.8% [28].

NETs are an uncommon type of tumor that originates from neuroendocrine cells, with the gastrointestinal tract and pancreas being the most common sites. In particular, rectal NETs account for approximately one-third of all gastrointestinal NETs [41, 42]. According to the mitotic count and the Ki-67 index, rectal NETs (≤1cm) are usually benign lesions [43]. However, as the size of the rectal NET increases, the risk of both lymph node and distant organ metastasis significantly increases [44]. In cases without vascular invasion, muscular invasion or lymph node metastasis, international guidelines recommend endoscopic resection for rectal NETs with a diameter of ≤1cm [45, 46]. Nonetheless, further investigation is still required to establish the efficacy of endoscopic treatment for rectal NETs. Recently, improved EMR has been shown to produce clinical outcomes that are as reliable as those achieved by ESD [30, 47].

A clinical study revealed that the en-bloc resection rate and histologically complete resection rate of rectal NETs treated with EMR were 99% and 72–74%, respectively; it was further considered that histologically complete resection might be related to tumor size, but not to endoscopic therapy [48]. However, our opinion was inconsistent regarding outcomes of endoscopic therapy. The Meta-analysis revealed that endoscopic therapy was closely related to local recurrence. The local recurrence rate of ESD was lower than that of EMR. We speculated that as long as the lesions did not invade the submucosa, the rate of en-bloc resection and local recurrence will gradually decrease with the improvement of endoscopic treatment. For skilled endoscopist, the size of the tumor may affect the operation time or postoperative complications. The complete resection rate of ESD was higher than that of EMR, which comparable to transanal endoscopic microsurgery [49]. Underwater EMR was superior to ESD in operation time, hospitalization time, operation cost, postoperative discomfort [50]. EMR with a ligation band device was more favorable for small rectal NETs (<10mm) with respect to clinical outcomes, procedure time, and technical difficulties [51]. Endoscopic resection was an effective treatment for small rectal NETs, while it still had a highly prevalent lymphovascular invasion in minute rectal NETs [52]. The findings of this Meta-analysis showed that ESD had a higher en bloc resection rate and R0 resection rate. Combined with the potential metastasis risk of rectal NETs, ESD may be the first choice and the optimal therapy for endoscopic resection.

Endoscopic resection is a minimally invasive treatment that can be used to treat early gastrointestinal cancer, such as esophageal cancer, gastric cancer, and colorectal cancer. It is important to achieve en-bloc and histologically complete resection during endoscopic treatment for a suspected or confirmed early colorectal cancer, according to the Clinical Practice Guideline [53].

EMR is an effective method for removing colorectal lesions that are smaller than 20 mm [54]. However, for larger lesions, piecemeal EMR may result in lower-quality and less reliable histopathologic findings, with higher rates of local recurrence and lower en-bloc resection rates [55]. In cases where colorectal lesions are highly suspected of being cancerous and limited to submucosal invasion, endoscopic submucosal dissection (ESD) may be considered for treatment [56]. ESD is widely considered a curative treatment for early colorectal cancer, as it achieves high rates of en-bloc resection and positive long-term outcomes for patients [57]. Compared to laparoscopic-assisted colorectal surgery, ESD produces lower complication rates and more favorable en-bloc and curative resection rates [58].

While ESD has a higher risk of delayed bleeding and perforation compared to EMR, these risks can be managed with certain precautions. Firstly, it is recommended that ESD be performed by experienced endoscopists. During the ESD procedure, high-frequency electrocoagulation can be used to stop bleeding at small vessels that are suspected of bleeding. Any bleeding points can also be identified and treated with high-frequency electrocoagulation after being clamped by hot forceps. In case of intraoperative perforation, metal clips can be applied to clamp the suspected or perforated sites for quick resolution of the issue.

One of the strengths of this Meta-analysis is the inclusion of a larger number of studies, including those from Chinese literature, which provides valuable insights from the latest clinical experience. Additionally, we used the GRADE tool to assess the quality of non-randomized controlled clinical trials, ensuring a more rigorous evaluation of the data. The heterogeneity of this Meta-analysis was only 60%, indicating a high level of consistency among the findings from each individual study. Our study explored various factors such as recurrence rate, en-bloc resection rate, R0 resection rate, operation time, and postoperative complications, which helped to explain the heterogeneity of results. Furthermore, we found that the operation mode had a significant impact on the recurrence rate, providing important information for improving treatment outcomes.

It is important to acknowledge that our research had several limitations. Firstly, there were differences in the lesion location in the selected literature, as the retrieval strategy used "colo-rectal," which may have excluded some rectal tumors. However, we tried to manually retrieve relevant studies to minimize this potential bias. Secondly, there were differences in the research design, inclusion criteria, and endoscopist's experience in diagnosing and removing colorectal tumors among the different studies, which was reflected in the high heterogeneity index. Lastly, the included studies were not all randomized, which may have weakened the evidence level of this Meta-analysis to some extent. It is important to consider these limitations when interpreting our findings.

## 5. Conclusions

Based on the findings of this Meta-analysis, it can be concluded that both EMR and ESD are viable options for the endoscopic treatment of colorectal tumors. However, ESD appears to have noticeable advantages over EMR in terms of en-bloc resection rate, R0 resection rate, and recurrence rate. Therefore, it is recommended to use ESD for the endoscopic treatment of colorectal tumors, if feasible. Nevertheless, it is important to note that more high-quality randomized controlled clinical trials are needed to further investigate the effectiveness of ESD in the treatment of colorectal cancer. These studies will help solidify the evidence base and potentially refine current treatment guidelines.

## Supporting information

**S1 Fig.** a. Complication rate after EMR. b. Complication rate after ESD.
(ZIP)

**S2 Fig.** a. Meta-regression analysis for recurrence rate of EMR, tumor size as co-variate. b. Meta-regression analysis for recurrence rate of ESD, tumor size as co-variate.
(ZIP)

**S1 Table. Result of meta regression analyses (Trowman method) to include tumor size as a co-variate and surgery time (EMR vs ESD) as the dependent variable.**
(DOCX)

**S1 Checklist. The meta-analysis was not registered and there was no protocol was prepared.**
(DOCX)

## Author Contributions

**Conceptualization:** Xinghuang Liu.

**Data curation:** Nian Wang, Lei Shu, Lin Yang.

**Investigation:** Zhaohong Shi.

**Methodology:** Xinghuang Liu.

**Project administration:** Tao Bai.

**Validation:** Song Liu, Lin Yang.

**Writing – original draft:** Nian Wang, Lei Shu.

**Writing – review & editing:** Song Liu, Zhaohong Shi.

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
