## [Decision Letter · Decision Letter 0]

11 Sep 2023

Endoscopic mucosal resection versus endoscopic submucosal dissection for colorectal adenomas and tumors: meta-analysis and systematic review

PONE-D-23-12685

Dear Dr. Xinghuang Liu

We’re pleased to inform you that your manuscript has been judged scientifically suitable for publication and will be formally accepted for publication once it meets all outstanding technical requirements.

Kind regards,

Paolo Aurello

Academic Editor

PLOS ONE

Reviewers' comments:

Reviewer's Responses to Questions

**Comments to the Author**

1. Is the manuscript technically sound, and do the data support the conclusions?

Reviewer #1: Yes

Reviewer #2: Yes

2. Has the statistical analysis been performed appropriately and rigorously? 

Reviewer #1: Yes

Reviewer #2: Yes

3. Have the authors made all data underlying the findings in their manuscript fully available?

Reviewer #1: Yes

Reviewer #2: Yes

4. Is the manuscript presented in an intelligible fashion and written in standard English?

Reviewer #1: Yes

Reviewer #2: Yes

5. Review Comments to the Author

Reviewer #1: The authors have reported their metanalysis comparing safety, efficacy and long-term outcome between endoscopic submucosal dissection (ESD) and endoscopic mucosal resection (EMR). The authors have reported their meta-analysis aiming to assess effectiveness and safety of ESD vs. EMR early rectal cancer. Unfortunately, only retrospective were included, with highly heterogeneous patient and surgical procedure characteristics. However, literature is clearly heterogenous and this paper show a new point of view on selection of patients.

Author could discussed the results in the western countries.

This subject is very interesting, and we need today to have some all results of therapeutic in order to limit impact of functional consequence of our strategies.

This paper was clear, and conclusion about this was strong.

Reviewer #2: interesting current work, the bibliography is rich recent in adequacy with the subject, this manuscript is to published , the data is important and the study is too . The results are interesting, satisfactory and correct, the conclusions are in line with the objectives of the study

6. PLOS authors have the option to publish the peer review history of their article (what does this mean?). If published, this will include your full peer review and any attached files.

Reviewer #1: No

Reviewer #2: **Yes: **Dr Ramzi GRAICHI

---

## [Editor Report · Acceptance letter]

19 Sep 2023

PONE-D-23-12685 

Comparing endoscopic mucosal resection with endoscopic submucosal dissection in colorectal adenoma and tumors: Meta-analysis and system review 

Dear Dr. Liu:

I'm pleased to inform you that your manuscript has been deemed suitable for publication in PLOS ONE. Congratulations! Your manuscript is now with our production department. 

Kind regards, 

on behalf of

Dr. Paolo Aurello 

Academic Editor

PLOS ONE